# Pillar[n]arene-Mimicking/Assisted/Participated Carbon Nanotube Materials

**DOI:** 10.3390/ma15176119

**Published:** 2022-09-03

**Authors:** Zhaona Liu, Bing Li, Zhizheng Li, Huacheng Zhang

**Affiliations:** 1Medical School, Xi’an Peihua University, Xi’an 710125, China; 2School of Chemical Engineering and Technology, Xi’an Jiaotong University, Xi’an 710049, China

**Keywords:** pillar[n]arene, carbon nanotube, synthesis, application, hybrid materials

## Abstract

The recent progress in pillar[n]arene-assisted/participated carbon nanotube hybrid materials were initially summarized and discussed. The molecular structure of pillar[n]arene could serve different roles in the fabrication of attractive carbon nanotube-based materials. Firstly, pillar[n]arene has the ability to provide the structural basis for enlarging the cylindrical pillar-like architecture by forming one-dimensional, rigid, tubular, oligomeric/polymeric structures with aromatic moieties as the linker, or forming spatially “closed”, channel-like, flexible structures by perfunctionalizing with peptides and with intramolecular hydrogen bonding. Interestingly, such pillar[n]arene-based carbon nanotube-resembling structures were used as porous materials for the adsorption and separation of gas and toxic pollutants, as well as for artificial water channels and membranes. In addition to the art of organic synthesis, self-assembly based on pillar[n]arene, such as self-assembled amphiphilic molecules, is also used to promote and control the dispersion behavior of carbon nanotubes in solution. Furthermore, functionalized pillar[n]arene derivatives integrated carbon nanotubes to prepare advanced hybrid materials through supramolecular interactions, which could also incorporate various compositions such as Ag and Au nanoparticles for catalysis and sensing.

## 1. Introduction

The investigation of macrocyclic compounds [1,2] had a very “Noble” beginning [3] with the study of crown ethers, known as the “first generation” of macrocyclic compounds [4,5,6], which awarded Cram [7], Lehn [8] and Pedersen [9] the Nobel Prize in Chemistry in 1987. Since then, a lot of marvelous research has been carried out to enrich the synthesis of different macrocyclic structures [10,11], as well as to expand the border of macrocyclic compound-based materials [12]. For example, macrocycle-modified, hybrid, multi-dimensional carbon materials [13,14] have attracted much more attention, due to their wide applications in biomedicine [15,16], catalysis [17], batteries [18] and sensors [17,19]. The most famous ones were macrocycle-based graphene [20]/carbon nanotube materials [21]. The physiochemical properties and behaviors of those carbon materials have been improved greatly in the presence of functionalized macrocyclic compounds [22]. However, the very interesting point of whether carbon materials such as carbon nanotubes ([Fig materials-15-06119-ch001]) could be directly prepared or not by a macrocyclic skeleton was rarely revealed [23].

Actually, macrocycles provided a dream structural skeleton as the missing piece for possible carbon materials such as carbon nanotubes via the art of organic synthesis [24,25,26]. Years ago [27], pillar[n]arene ([Fig materials-15-06119-ch001]) was reported by Ogoshi et al. as the “fifth generation” of macrocyclic compounds [28,29,30,31,32], which was composed of hydroquinone units that were bridged by methylene subunits at para-positions, leading to the fabrication of an electron-rich cavity [33,34], pillar-shaped cyclic rigid molecular architectures [35], and unique planar chirality [36,37,38,39]. The first thing needed for utilizing pillar[n]arene from the organic synthesis view was to construct another famous carbon material: the carbon nanotube [40,41]. Up to now, several classic, organic syntheses [26] have been employed to mimic the structure of carbon nanotubes by using the skeleton of pillar[n]arene, leading to interesting porous materials [42]. Additionally, the proper functionalization of the “short” cylindrical pillar[n]arene can also be similar to the characteristics of carbon nanotubes. Due to the introduction of the architecture of functionalized pillar[n]arene as well as supramolecular interactions, the physiochemical properties of modified pillar[n]arene were very similar to those of carbon nanotubes, resulting in a lot of work being carried out to compare the diverse performance of pillar[n]arene and carbon nanotubes in several fields, such as artificial channels [43]. Furthermore, functional pillar[n]arene derivatives, such as the water-soluble one, were used as an amphiphilic addictive [44] to promote the physiochemical properties of carbon nanotubes, e.g., by greatly improving the dispersion of carbon nanotubes in aqueous solutions. Finally, functional pillar[n]arene was directly decorated on the surface of carbon nanotubes to expand the current applications, constructing pillar[n]arene-carbon nanotube hybrid materials for sensing [45], catalysis [46] and supramolecular gels [47].

In this review, we initially summarize the pillar[n]arene-assisted carbon nanotube materials. At the very beginning, the structural skeleton of pillar[n]arene was taken as the basis for synthesizing linear, rigid, oligomeric/polymeric and flexible, conformational, peptide-modified, advanced, one-dimensional and spatially limited architectures to resemble the morphological molecular structure and physiochemical characteristics of carbon nanotubes, leading to various interesting applications such as the adsorption of toxic pollutants, gas adsorption and separation, as well as artificial water channels. In addition, water-soluble pillar[n]arene could include hydrophobic guest molecules into self-assembled amphiphiles to assist in dispersing carbon nanotubes in aqueous solutions. Furthermore, both polymeric and water-soluble pillar[n]arene have the capacity of integrating carbon nanotubes in the absence and presence of other inorganic compositions, respectively, for the preparation of carbon nanotube-based hybrid materials, paving the way for supramolecular organogels, catalysis and sensing. Finally, we also discuss new challenges in the Overview and Outlook section, and provide primary suggestions for future work in this field.

## 2. Mimicking the Structure and Characteristics of Carbon Nanotubes by Functionalized Pillar[n]arene

### 2.1. Preparing Linear Pillar[n]arene-Based Oligomer/Polymer via Rigid Aromatic Bridges

Due to the possession of rigid pillar-like molecular structures and an electron-rich cavity [48], the skeleton of functionalized pillar[n]arene was employed as previous pieces for the construction of linear oligomeric and polymeric architectures, such as **P1**–**P7** (Figure 1), by introducing rigid aromatic bridging subunits to mimic carbon nanotubes [42,49,50,51,52]. Several classic organic syntheses such as heterocyclization and Pd-catalyzed coupling reactions [53,54,55,56] have been thoroughly employed. Particularly, those carbon nanotube-resembling linear pillar[n]arene-based porous materials have been used in diverse applications such as gas absorption [57,58], as well as the absorption and separation of toxic pollutants in water [59]. For example, **P2** exhibits a recognition towards solvent molecules such as dichloromethane [50], whereas **P3** has the ability to capture toxic pollutants such as adiponitrile [51]. In addition, **P5** can selectively absorb CO_2_ rather than N_2_ or methane [52], whereas **P6** and **P7** were used for separating propane gas from the simulated gas mixture of methane and propane [42].

### 2.2. Preparing Peptide-Appended Pillar[n]arene Processing Intramolecular Hydrogen Bonds

Except for introducing rigid aromatic moieties as the bridging linker for the construction of pillar[n]arene-based polymeric architectures, diverse designs and synthetic strategies were carried out for mimicking carbon nanotubes; for example [60], the peptide-appended pillar[n]arene **P8** (Figure 2) was produced to introduce intramolecular interactions such as hydrogen bonding [61] to form “closed”, tubular, molecular architectures [62], as well as to resemble the performance of carbon nanotubes as artificial water channels [63,64] and permeable membranes [65,66,67]. It was revealed that the average single-channel osmotic water permeability [65] and the ion rejection of **P8** were greatly analogous to those of carbon nanotubes [67]. Furthermore, the pore density [68] of **P8**-based channel arrays was much higher than that of carbon nanotube-based ones [65]. It was also found that the flexible conformation of the peptide-appended pillar[n]arene was available for water permeability [66,69,70].

## 3. Dispersion of Carbon Nanotube by Using Functionalized Pillar[n]arene

Water-soluble pillar[n]arene could include hydrophobic guest molecules [71,72], producing pillar[n]arene-based self-assembled amphiphiles (PSAs) [73,74] and resembling the performance of general surfactant-dispersing carbon nanotubes [75]. For example [76], water-soluble carboxylate-perfunctionalized [77] pillar[6]arene **P9** (Figure 3 and Table 1) has the ability to recognize the hydrophobic pyrene [78] derivative **G1** (Figure 3) in the stoichiometry of 1/1, and aggregate into vesicular architectures [79] in an aqueous solution as confirmed by transmission electron microscopy (TEM) [80]. Since the water solubility of **P9** changes with the change of pH value, the morphology of **P9** ⸧ **G1**-based self-assemblies could be also controlled. In addition [76], **P9** could include another pyrene derivative **G2** (Figure 3) with the association constant (*K_a_*) of (8.04 ± 0.68) 10^4^ M^−1^. Additionally, the obtained amphiphilic inclusion could further disperse multi-walled carbon nanotubes well by sonication in aqueous solutions (Figure 1 and Table 1) as confirmed by TEM and scanning electron microscopy (SEM). The π–π stacking interactions [81] between pyrene subunits and carbon nanotubes played a significant role in this process as confirmed by fluorescence spectroscopy [82].

Similarly [83], another pyrene derivative **G3** (Figure 3 and Table 1) which is responsive to UV irradiation and degrades into **F1** and **F2** (Figure 3) was further employed as the hydrophobic guest to be included by **P9**. Thus, the amphiphilic inclusion **P9** ⸧ **G3** was obtained and exhibited the critical aggregation concentration (CAC) [84] of 1.0 × 10^6^ mol L^−1^, which has the capacity of dispersing multi-walled carbon nanotubes in aqueous solutions as confirmed by TEM (Table 1). Interestingly, the dispersion of carbon nanotubes could be controlled upon UV irradiation according to the formation/deformation of **G3**.

**Table 1 materials-15-06119-t001:** Fabrication of carbon nanotube-based hybrid materials by using different functional pillar[n]arene, other diverse compositions, various guests for applications such as gels, sensing and catalysis.

Pillar[n]arene	Other Composition	Hybrid Material	Guest	Application	Ref.
**P9**	-	-	**G1** and **G2**	Amphiphilic **P9** ⸧ **G2** dispersing carbon nanotube in aqueous solution	[76]
**P9**	-	-	**G3** which could be degraded into **F1** and **F2** upon UV irradiation	Amphiphilic **P9** ⸧ **G3** dispersing carbon nanotube in aqueous solution	[83]
**P10** and **P11**	**F3**	(**P11** ⸧ **G4**) non-covalently complexing carbon nanotube	**G4**	Supramolecular gels	[85]
**P12**	Ag nanoparticles	Ag@(**P12** non-covalently interacting with carbon nanotube)	**G5**–**G7**	Catalysis and sensing	[86]
**P13**	Ag nanoparticles	Au@(**P13** non-covalently interacting with carbon nanotube)	**G8**	Catalysis and sensing	[87]

## 4. Decoration of Carbon Nanotube by Using Functional Pillar[n]arene

The mono-hydroxyl functionalized pillar[5]arene **P10** (Figure 4) coupled with the alkyl bromide-modified polyfluorene **F3** (Figure 4) via a classic nucleophilic substitution reaction led to the formation of pillar[5]arene-functionalized polymer **P11** (Figure 4 and Table 1) [85]. Due to the possession of the pillar[5]arene cavity on the structural skeleton, **P11** could form a complex with a neutral poly(ethylene glycol) (PEG) derivative with two ending subunits—alkylnitrile **G4** (Figure 4) via donor–acceptor interactions. Particularly, this obtained host–guest inclusion **P11** ⸧ **G4** not only has the capacity for properly exfoliating and dispersing single-walled carbon nanotubes in organic solvents (600 μg mL^−1^, Figure 2) as confirmed by ^1^H NMR, Raman, UV–Vis–near-infrared (NIR) spectra, and thermogravimetric analysis (TGA), but also affords the preparation of pillar[5]arene-based polymer-carbon nanotube composite organogels [88,89,90] in 1,2-dichlorobenzene (40 *wt*%) via non-covalent interactions (Figure 3) [85].

Besides the organic and polymeric composition, diverse inorganic materials were also employed for fabricating different functional, carbon nanotube-based, hybrid materials. For example [86], the water-soluble phosphate-perfunctionalized pillar[6]arene **P12** (Figure 5 and Table 1) could decorate the surface of a single-walled carbon nanotube at room temperature via π–π stacking interactions between the carbon nanotube and benzene moieties of **P12** by sonicating in aqueous solutions, as confirmed by zeta potentials and Fourier transform infrared (FTIR) spectroscopy. Particularly, Ag nanoparticles could be further well dispersed on the surface of the carbon nanotube due to the coordination environment provided by the cavity of **P12** (Figure 4 and Figure 5). Thus, the obtained hybrid materials containing Ag nanoparticles, carbon nanotubes and **P12** (Figure 4 and Figure 5 and Table 1) exhibited strong catalytic activity towards a series of guest molecules such as 4-nitrophenol (**G5**, Figure 5), methylene (**G6**, Figure 5) and paraquat (**G7**, Figure 5), paving the way for efficient electrochemical sensing of highly toxic herbicides.

Except for loading Ag nanoparticles, Au nanoparticles [91,92,93] could also be introduced into carbon nanotube-based hybrid materials via the assistance of pillar[n]arene. For example, water-soluble hydroxyl pillar[5]arene **P13** (Figure 5) was also used for dispersing single-walled carbon nanotubes in aqueous solutions via non-covalent interactions, and further assisted in promoting the formation of Au nanoparticles on the surface of carbon nanotubes, leading to the formation of hybrid materials Au@(P13 interacting with carbon nanotubes) (Table 1) [87]. It has been revealed that such hybrid materials had reasonable performances in catalyzing an ethanol oxidation reaction (EOR), as well as sensing *p*-dinitrobenzene (**G8**, Figure 5) because of pillar[5]arene cavities.

## 5. Overview and Outlook

In conclusion, we summarized the recent progress of pillar[n]arene-assisted carbon nanotube hybrid materials. During the preparation of such hybrid materials, pillar[n]arene could initially play different, but significant, roles. For example, either aromatic linkers or peptide subunits could be introduced during the syntheses to transfer pillar[n]arene moieties into carbon nanotube-like molecular structures and physiochemical characteristics, leading to the formation of one-dimensional rigid oligomers and polymers by covalent bonds, as well as spatially limited, “closed”, artificial water channels containing intermolecular interactions. Particularly, by taking advantage of the art of organic synthesis, those carbon nanotube-resembled molecular architectures exhibited reasonable performances in applications of gas adsorption, adsorption and separation of toxic pollutants, as well as artificial channels and membranes with ion rejection. Furthermore, due to the formation of self-assembled amphiphiles by including hydrophobic neutral and positive-charged guest molecules in aqueous solutions, water-soluble pillar[n]arene could greatly improve and control the dispersion of carbon nanotubes in accordance with different external stimuli such as changed pH values. Finally, functional pillar[n]arene derivatives could non-covalently integrate carbon nanotubes into diverse, advanced, hybrid materials in the absence/presence of other inorganic compositions such as Ag and Au nanoparticles, revealing attractive activity in catalysis and sensing.

A lot of perspective work in this area is still attractive to researchers, for example, in exploring synthesis methods and designing more functional materials.

Firstly, more functional pillar[n]arene derivatives could be introduced to assist and participate in the fabrication of carbon nanotube-based hybrid materials. For example, up to now, only pillar[5]arene and pillar[6]arene were employed in this field, and other larger-sized pillar[n]arene was not used (Figure 6) [94]. As known, pillar[n]arene with bigger cavities not only exhibits a different molecular geometry and shape [95,96], but also shows different behaviors in physiochemical characteristics such as host–guest inclusions [97,98]. Thus, the encapsulation process of pillar[n]arene-based carbon nanotube hybrid materials should be further explored.

Secondly, more efforts should be made to investigate the new fabrication methods of pillar[n]arene-involved carbon nanotube-based hybrid materials; for example, how can we build hybrid materials by covalently coupling pillar[n]arene moieties and carbon nanotubes together? Covalent bonds may promote the stability of this hybrid material, but provide great challenges in employing/choosing proper organic synthesis strategies. Additionally, particular carbon materials such as carbon ends saturated with hydrogen atoms in carbon nanotubes, as well as different sizes of carbon nanotubes should also be employed in future research for exploring new construction strategies.

Thirdly, pillar[n]arene-assisted carbon nanotubes have shown a very good example for exploring novel pillar[n]arene-assisted carbon materials such as pillar[n]arene-based fullerene [99]/carbon black/C_3_N_4_-containing hybrid materials [100]. The expansion of those studies will not only enhance the physiochemical feature generated from carbon nanotube-based hybrid materials, [101,102,103] but also greatly enrich the current family of hybrid carbon materials, paving the way to possibly enlarge the fields of attractive applications [104].

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
