# Peer review of "Pillar[n]arene-Mimicking/Assisted/Participated Carbon Nanotube Materials"

_materials, 2022, doi:10.3390/ma15176119_

Round 1

Reviewer 1 Report

This review aims to provide an overview of pillare[n]arene CNT composite materials. The author clearly have extensive knowledge on pillare[n]arene-based materials as shown in several reviews (Refs. 20, 48 and 73 in the manuscript) as well as research articles. However, the current manuscript has failed to present a clear overview on such topic largely due to how it is presented.  I suggest the authors take the trouble to rewrite this review.

1. The topic of this review paper changed in the middle. At the beginning it was about pillare[n]arene modified hybrid materials, but immediately after the introduction, the authors started talking about using pillare[n]arene to mimick the function of carbon nanotube. In my view, the section 2 has nothing to do with the title and topic of this review. But if this section is taken out, the whole review seems to not carry much weight. 

2. This review is obviously not in-depth enough, even though the number of citation is large, the entire article is limited in length and the author have not put enough discussion and their own views. What makes this topic attractive? How will this area develop in the future?

3. The readability of the article is not high and the wording is often not ideal. 

4. What ref. 63 has described is not a pillare[n]arene, but a hybrid[4]arene. 

Author Response

To Reviewer 1

This review aims to provide an overview of pillare[n]arene CNT composite materials. The author clearly have extensive knowledge on pillare[n]arene-based materials as shown in several reviews (Refs. 20, 48 and 73 in the manuscript) as well as research articles. However, the current manuscript has failed to present a clear overview on such topic largely due to how it is presented.  I suggest the authors take the trouble to rewrite this review.

Thanks for the reviewer’s positive comments on our manuscript. We have revised the manuscript according to her/his valuable suggestions. All revised sections have been highlighted in yellow color. We do hope that the reviewer will be satisfied with our efforts.

  1. The topic of this review paper changed in the middle. At the beginning it was about pillare[n]arene modified hybrid materials, but immediately after the introduction, the authors started talking about using pillare[n]arene to mimick the function of carbon nanotube. In my view, the section 2 has nothing to do with the title and topic of this review. But if this section is taken out, the whole review seems to not carry much weight.

To be honest, there were too many research papers about pillar[n]arene-based hybrid materials, which look like to be composed by any “reported” organic and inorganic compounds and materials. In our opinion, it might not be good enough for researchers to design the ideal hybrid composites just by physically/chemically mixing/coupling pillar[n]arene and other materials/compounds together. There should be some “good” reason to do it, or we should know something valuable “behind the scene”. For example, like the hybrid materials composed of pillar[n]arene and carbon nanotube, there is an interesting relationship between them. The molecular architecture of pillar[n]arene could act as the building block in synthesis for mimicking carbon nanotube, where the cavity of pillar[n]arene could be enlarged by proper functionalization. That is also a good reason to distinguish pillar[n]arene from other valuable cyclophanes such as calix[n]arene. Thus, in the section about “Mimicking the structure and characteristics of carbon nanotube by functionalized pillar[n]arene”, we mainly discussed the art of organic synthesis to modify pillar[n]arene for the fabrication of carbon nanotube-like molecular/polymeric architectures. In our opinion, it is important for readers to understand both structures and physical characteristics of pillar[n]arene and carbon nanotube. And it is helpful to build the reasonable bridge between those two composites.

  1. This review is obviously not in-depth enough, even though the number of citation is large, the entire article is limited in length and the author have not put enough discussion and their own views. What makes this topic attractive? How will this area develop in the future?

The organization and presentation of this review are to address attentions from researchers in supramolecular chemistry, organic synthesis chemistry, physical organic chemistry and material science. We believe the categorization of different sections in this review such as “Mimicking the structure and characteristics of carbon nanotube by functionalized pillar[n]arene”, “Dispersion of carbon nanotube by using functionalized pillar[n]arene”, and “Decoration of carbon nanotube by using functional pillar[n]arene” were proposed and summarized by ourselves for the first time. Also, we would like to initially address the attention of using pillar[n]arene to mimic and functionalize carbon nanotube. But due to the under development of this field, a lot of work need to be done in the future, and we have prosed several issues in the conclusion section. More references and discussions have been added in the revised conclusion section. 

  1. The readability of the article is not high and the wording is often not ideal.

We have revised the manuscript according to the reviewer’s valuable suggestions.

  1. What ref. 63 has described is not a pillare[n]arene, but a hybrid[4]arene.

This reference is to help readers to understand the concept of “artificial water channel”.

Reviewer 2 Report

Dear Authors

This review is focused on the recent progress in pillar[n]arene-assisted/participated carbon nanotube hybrid materials. The following suggestion and comments should be taken:

1. The authors could insert more numerical data into the Abstract for enhancement of the manuscript.

2. The overall English needs to be improved. Please seek guidance from a native English speaker if possible ("the" "a", commas, plural form and others could be corrected).

3. The introduction section needs enhancement 1-3 sentences about possible hybrid materials with carbon nanotubes. Please cite (1) Materials 2013, 6(6), 2534-2542; https://doi.org/10.3390/ma6062534 (2) Materials 2021, 14(9), 2448; https://doi.org/10.3390/ma14092448 (3) Microsyst Technol 2021 https://doi.org/10.1007/s00542-021-05211-6

4. Scheme 5. Please change for better quality.

5. Please answer the question in the comments: What about the encapsulation process in tubular structures? Which model is more favourable?

6. What about carbon ends saturated with hydrogen atoms in carbon nanotubes? Please explain it in the comments.

7. What about the dependence on the structure size of nanotubes?

8. Could the authors add some results of elemental analysis or XPS from other articles?

9. Authors are suggested to describe some applications and future plans in conclusions.

10. Please see the structure of the manuscript for Materials and please correct the manuscript.

11. Please add the section: Author Contributions.

Author Response

To Reviewer 2

This review is focused on the recent progress in pillar[n]arene-assisted/participated carbon nanotube hybrid materials.

Thanks for the reviewer’s valuable comments on our review manuscript. We have revised the manuscript according to her/his suggestions. All revised sections have been highlighted in yellow color. We do hope that the reviewer will be satisfied with our efforts.

The following suggestion and comments should be taken:

  1. The authors could insert more numerical data into the Abstract for enhancement of the manuscript.

Because this is a review and we summarized a lot of research papers in the manuscript, it will be difficult for us to only address one or two exact/accurate “data” in the abstract section. We suggest that the abstract section is try to show readers the complete picture in this section.

  1. The overall English needs to be improved. Please seek guidance from a native English speaker if possible ("the" "a", commas, plural form and others could be corrected).

We have revised the manuscript according to the reviewer’s valuable suggestions.

  1. The introduction section needs enhancement 1-3 sentences about possible hybrid materials with carbon nanotubes. Please cite (1) Materials 2013, 6(6), 2534-2542; https://doi.org/10.3390/ma6062534 (2) Materials 2021, 14(9), 2448; https://doi.org/10.3390/ma14092448 (3) Microsyst Technol 2021 https://doi.org/10.1007/s00542-021-05211-6

We have added those valuable papers as reference 101-103 in the conclusion section, in which we propose a future research direction for this hybrid material.

  1. Ordoñez-Casanova, E.G.; Román-Aguirre, M.; Aguilar-Elguezabal, A.; Espinosa-Magaña, F. Synthesis of Carbon Nanotubes of Few Walls Using Aliphatic Alcohols as a Carbon Source. Materials 2013, 6, 2534-2542.
  2. Kamedulski, P.; Lukaszewicz, J.P.; Witczak, L.; Szroeder, P.; Ziolkowski, P. The Importance of Structural Factors for the Electrochemical Performance of Graphene/Carbon Nanotube/Melamine Powders towards the Catalytic Activity of Oxygen Reduction Reaction. Materials 2021, 14, 2448.
  3. Shoukat, R.; Khan, M.I. Carbon nanotubes: a review on properties, synthesis methods and applications in micro and nanotechnology. Microsyst Technol. 2021, 27, 4183-4192.

  1. Scheme 5. Please change for better quality.

It has been revised in the resubmission.

  1. Please answer the question in the comments: What about the encapsulation process in tubular structures? Which model is more favourable?

We were not sure about which tubular structure the reviewer was talking about. Actually, if he/she mentioned the pillar[n]arene-resembled carbon nanotube in section 2, the cavity of pillar[n]arene played a significant role in the encapsulating process via named “host-guest interaction”, which we have discussed in the review. In addition, we have added this point in the conclusion section for calling attentions of researchers in future studies. 

  1. What about carbon ends saturated with hydrogen atoms in carbon nanotubes? Please explain it in the comments.

We were not sure the meaning of this question. If the reviewer is talking about the modification of carbon nanotube by using pillar[n]arene, the current modification method employed in the reported cases were mainly based on named π—π stacking interactions. Thus, we proposed that other modification methods should be developed based on diverse cases of carbon nanotube in the conclusion section. In addition, we have added this point in the conclusion section for calling attentions of researchers in future studies. 

  1. What about the dependence on the structure size of nanotubes?

Current researches did not focus on this point, but we have added this issue in the conclusion section for calling attentions of researchers in future studies. 

  1. Could the authors add some results of elemental analysis or XPS from other articles?

Elemental analysis and XPS were not the focus in our review, and we mainly focused on the design and preparation strategy of fabricating hybrid materials. Thus, we did not provide those data in this review.

  1. Authors are suggested to describe some applications and future plans in conclusions.

We have discussed several issues in this field, and proposed future work in the conclusion section. To be honest, the application in current researches was limited, and we have called attentions in this field in the conclusion section.

  1. Please see the structure of the manuscript for Materials and please correct the manuscript.

We have revised it accordingly.

  1. Please add the section: Author Contributions.

We have added a new section to address this issue.

Round 2

Reviewer 1 Report

After seeing the author's second draft and response, I have to stick with my original opinion.

Second part may be important, but I still don't see it have anything to do with the title. Perhaps the authors should come up with a new title other than the current one.

Regarding the fourth answer, there are so many examples of using pillararene as artificial channels, I do not see the reason to cite a work of other molecules. This paper should be focused on pillararene, not other cyclophanes. And also the mechanism of hybrid[4]arene worked as AWC is completely different from pillararene. Citing this paper the way the author did brings confusion. 

What makes pillararene/CNT hybrid materials worth exploring? Are there any  characteristics and potential of this hybrid materials that other materials or specifically, other cyclophane/CNTs materials cannot compete?

Line 13, what does perfunctionalizing mean? 

Overall, I am not convinced by the authors that this manuscript is suitable for publication in Materials.

Author Response

To Reviewer 1

After seeing the author's second draft and response, I have to stick with my original opinion.

Second part may be important, but I still don't see it have anything to do with the title. Perhaps the authors should come up with a new title other than the current one.

Reply: We have changed the title into “Pillar[n]arene-mimicking/assisted/participated Carbon Nano-tube Materials” as suggested by the reviewer.

Regarding the fourth answer, there are so many examples of using pillararene as artificial channels, I do not see the reason to cite a work of other molecules. This paper should be focused on pillararene, not other cyclophanes. And also the mechanism of hybrid[4]arene worked as AWC is completely different from pillararene. Citing this paper the way the author did brings confusion.

Reply: We have changed the reference according to the reviewer’s suggestion.

What makes pillararene/CNT hybrid materials worth exploring? Are there any  characteristics and potential of this hybrid materials that other materials or specifically, other cyclophane/CNTs materials cannot compete?

We have explained this issue in the introduction section, cover letter and the previous point-to-point repoint letter. The explanation in the previous response letter was shown as following,

To be honest, there were too many research papers about pillar[n]arene-based hybrid materials, which look like to be composed by any “reported” organic and inorganic compounds and materials. In our opinion, it might not be good enough for researchers to design the ideal hybrid composites just by physically/chemically mixing/coupling pillar[n]arene and other materials/compounds together. There should be some “good” reason to do it, or we should know something valuable “behind the scene”. For example, like the hybrid materials composed of pillar[n]arene and carbon nanotube, there is an interesting relationship between them. The molecular architecture of pillar[n]arene could act as the building block in synthesis for mimicking carbon nanotube, where the cavity of pillar[n]arene could be enlarged by proper functionalization. That is also a good reason to distinguish pillar[n]arene from other valuable cyclophanes such as calix[n]arene. Thus, in the section about “Mimicking the structure and characteristics of carbon nanotube by functionalized pillar[n]arene”, we mainly discussed the art of organic synthesis to modify pillar[n]arene for the fabrication of carbon nanotube-like molecular/polymeric architectures. In our opinion, it is important for readers to understand both structures and physical characteristics of pillar[n]arene and carbon nanotube. And it is helpful to build the reasonable bridge between those two composites.

Line 13, what does perfunctionalizing mean?

The concept of “perfunctionalizing” could be found in the reference of “Nathan L. Strutt, Huacheng Zhang, Severin T. Schneebeli, and J. Fraser Stoddart. Accounts of Chemical Research 2014 47 (8), 2631-2642. DOI: 10.1021/ar500177d”. And it is popular and accepted in pillararene chemistry.

Overall, I am not convinced by the authors that this manuscript is suitable for publication in Materials.

We are sorry for this. We have labeled all the new revision section in green color. We have tried our best, and we do hope that the reviewer will be satisfied with our efforts during such tough time of COVID-19.

Reviewer 2 Report

The authors have addressed all comments and the manuscript can be published as is.

Author Response

Thanks a lot for the support from this reviewer. We do appreciate his/her efforts and time.
